# Design of an inherently-stable water oxidation catalyst

Biswarup Chakraborty[1], Gal Gan-Or[1], Manoj Raula[1], Eyal Gadot[1] & Ira A. Weinstock [1]

While molecular water-oxidation catalysts are remarkably rapid, oxidative and hydrolytic processes in water can convert their active transition metals to colloidal metal oxides or hydroxides that, while quite reactive, are insoluble or susceptible to precipitation. In response, we propose using oxidatively-inert ligands to harness the metal oxides themselves. This approach is demonstrated by covalently attaching entirely inorganic oxo-donor ligands (polyoxometalates) to 3-nm hematite cores, giving soluble anionic structures, highly resistant to aggregation, yet thermodynamically stable to oxidation and hydrolysis. Using orthoperiodate (at pH 8), and no added photosensitizers, the hematite-core complex catalyzes visible-light driven water oxidation for seven days (7600 turnovers) with no decrease in activity, far exceeding the documented lifetimes of molecular catalysts under turnover conditions in water. As such, a fundamental limitation of molecular complexes is entirely bypassed by using coordination chemistry to harness a transition-metal oxide as the reactive center of an inherently stable, homogeneous water-oxidation catalyst.

[1] Department of Chemistry and the Ilse Katz Institute for Nanoscale Science & Technology, Ben-Gurion University of the Negev, Beer Sheva 84105, Israel. Correspondence and requests for materials should be addressed to I.A.W. (email: iraw@bgu.ac.il)

A current challenge in the development of molecular water-oxidation catalysts[1] is to overcome their inherent susceptibilities to oxidative or hydrolytic degradation under turnover conditions in water[2,3]. Advances in the past decade have led to remarkably fast rates of $O_2$ formation[4–6], along with the parallel development of more stable catalysts[7], better able to withstand strongly oxidizing conditions in water. Nevertheless, organic ligands are susceptible to oxidation[1,2,8–11], which, combined with hydrolysis of their active transition-metal centers, often results in the formation of colloidal metal oxides. Entirely inorganic polyoxometalate (POM)-based water-oxidation catalysts[12,13], by contrast, are thermodynamically stable to oxidation[12–15], can utilize earth-abundant metals[5,14–17], and feature remarkably rapid rates[5]. At the same time, molecular POM catalysts are stable to hydrolysis only under specific, albeit well-defined and by now, well-understood conditions, concerning variables such as pH and the nature and concentration of buffer[17–19]. Fundamentally, however, their catalytically active transition metals are in equilibrium with trace concentrations in the aqueous solvent[20], which can lead to hydrolysis under non-optimal conditions[18]. Hence, as advances in ligand design have led to impressively rapid rates and longer catalyst lifetimes under turnover conditions, catalyst stability nevertheless remains an ongoing topic of discussion and experiment[1,10,20].

One reason for this heightened level of concern is that colloidal metal oxides can be extremely active[21–23], such that considerable efforts are required to prove that the molecular catalyst, and not its products of decomposition in water, are the kinetically competent species in catalytic water oxidation. Given this situation, a compelling solution to catalyst stability might be to embrace the colloidal metal-oxide nanocrystals (NCs) themselves as thermodynamically stable water-oxidation catalysts. Apart from a few exceptions[23], however, colloidal metal oxides are either insoluble at most pH values in water, or rapidly aggregate and precipitate from water under turnover conditions. And, despite impressive advances in solvothermal syntheses of metal-oxide NCs[24], the requisite organic stabilizing ligands are not only susceptible to oxidation, but usually limit solubility in water and block access to the metal-oxide surface.

In this context, we recently discovered that heteropolytungstate cluster anions (POMs) could serve as covalently attached oxo-donor ligands for 6-nm anatase-$TiO_2$ NCs[25], giving water-soluble POM-complexed nanostructures. We now deploy Fe(III)-substituted $[\alpha\text{-}P^{V}W^{VI}_{11}O_{39}]^{7-}$ cluster anions as oxo-donor ligands for hematite ($\alpha\text{-}Fe_2O_3$) cores, giving water-soluble catalysts, **1**, uniquely positioned between molecular iron-oxide clusters[26] and colloidal hematite[27,28]. The new catalyst features 3-nm hematite ($\alpha\text{-}Fe_2O_3$) cores comprised of ca. 300 Fe atoms, sufficiently large to possess the visible-light semiconductor properties of hematite. Unlike colloidal hematite, however, covalent coordination of each $\alpha\text{-}Fe_2O_3$ core by ca. 15 cluster anions renders the POM-complexed structures stable to aggregation in water, giving optically transparent solutions at pH values of 2.5 to 8. Moreover, fitted with entirely inorganic tungsten(VI)-oxide ligands, and formed in water at 220 °C, **1** is thermodynamically stable to both oxidative degradation and hydrolysis. As an inherently stable visible-light activated water-oxidation catalyst, **1** is capable of continuous operation for 7 days under turnover conditions (corresponding to 7600 turnovers) with no decrease in activity.

## Results

**Inorganic coordination complexes of hematite**. Complex **1** is prepared by converting micron-sized particles of $\gamma\text{-}FeO(OH)$ to $\alpha\text{-}Fe_2O_3$ (at 220 °C) in the presence of $[\alpha\text{-}PW_{11}O_{39}]^{7-}$, after which, numerous Fe(III)-substituted anions, $[\alpha\text{-}PW_{11}O_{39}Fe^{III}]^{4-}\text{–}O^-$,

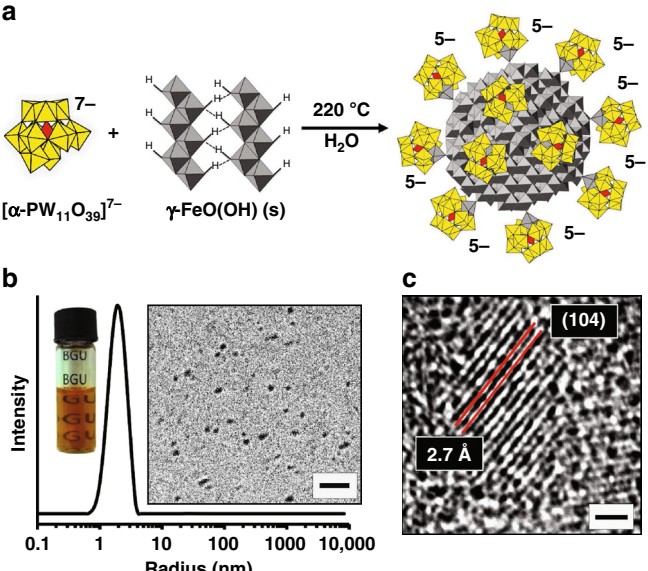

**Fig. 1** Synthesis of the cluster-anion ligated hematite-core complex, **1**. **a** Synthesis of **1** by reaction of partially amorphous $\gamma\text{-}FeO(OH)$ with $\alpha\text{-}PW_{11}O_{39}^{7-}$ at 220 °C at pH 6.5 in water. An illustration of **1** is shown at the right; gray: Fe(III)-centered polyhedra; yellow: W(VI)-centered polyhedra; red: P(V)-centered tetrahedra. **b** Number-weighted dynamic light scattering of a clear-orange solution of **1** indicated particles with an average hydrodynamic radius of 1.9 nm. The left inset is a photograph of the clear-orange solution; the right inset is a cryo-TEM image of freely soluble particles of **1** in rapidly vitrified water (scale bar: 10 nm). **c** HRTEM image of the hematite core of **1** with fringes corresponding to (104) planes (scale bar: 1 nm)

remain bound via bridging oxo linkages to 3-nm hematite cores (Fig. 1a and Supplementary Figures 1-7). The reaction gives an orange, optically transparent pH-7 solution of **1** (Fig. 1b, left inset). Dynamic light scattering (DLS) of the clear-orange solution reveals a number-weighted hydrodynamic radius of 1.9 nm (Fig. 1b). Cryogenic-TEM images[29] of the same (vitrified) solution (right inset to Fig. 1b) reveal numerous freely diffusing particles with an average size of 2.8–3.0 nm, too small for resolution of their surface structures in cryo-TEM images (additional images in Supplementary Figure 8). (This limitation is consistent with extensive studies of POMs on gold nanoparticles, in which even well-ordered POM monolayers are not discernible on gold cores smaller than ca. 5 nm[29].)

The cores of **1** are single NCs of hematite ($\alpha\text{-}Fe_2O_3$). High-resolution TEM images (Fig. 1c) reveal an inter-planar spacing of 2.69 Å, in agreement with the (104) crystal planes of hematite[30]. Indexing of the well-defined rings found in electron diffraction of dry samples precisely matches hematite (Supplementary Figure 9a)[31], and the related dark-field images reveal individual 2.8 ± 0.5 nm (± denotes s.d.) diameter NCs (Supplementary Figure 10). The hematite structure was further confirmed by powder X-ray diffraction (XRD)[30], for which the Debye–Scherrer equation[32] gave a crystallite size of 3.5 ± 0.5 nm (Supplementary Figures 9b and 11).

**Isolation and stability to aggregation**. To isolate pure samples of **1**, a clear-orange solution (inset to Fig. 1b) was made 2 M in NaCl. This gave a cloudy solution of salted-out **1**, an orange-red solid isolated by centrifugation. Re-dissolution of **1** in pure water (10 mL) returned clear-orange solutions, even after multiple cycles of NaCl addition, centrifugation, and re-dissolution (Supplementary Figure 2). This solubility and remarkable stability to

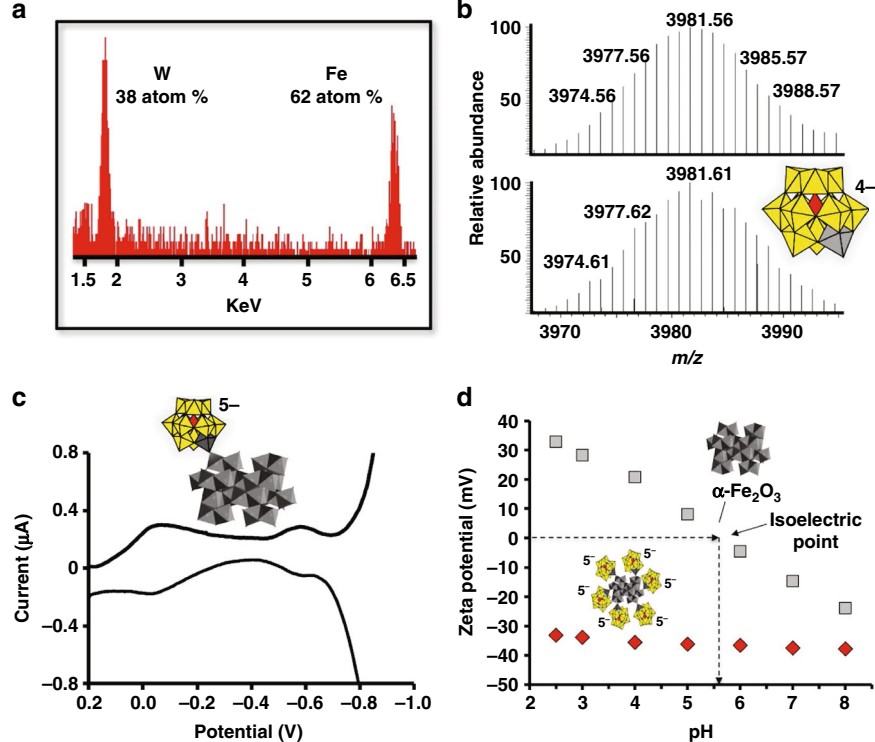

**Fig. 2** Characterization of cluster-anion ligands on the hematite cores of **1**. **a** EDX spectrum of **1**. **b** Top: ESI-MS spectrum; bottom: a simulated spectrum, showing a precise match with the Keggin ion [(TBA)$_5${PW$_{11}$O$_{39}$Fe$^{III}$(H$_2$O)$_2$}]$^+$ (see Supplementary Figure 14). Inset: the Keggin anion, with the Fe(III)-centered polyhedron in gray. (In water, the outward-facing coordination site on Fe(III) is occupied by H$_2$O or OH$^-$, or by a $\mu_2$-O$^{2-}$ linkage to Fe(III) in a second cluster anion.). **c** Differential pulse voltammogram (DPV) of **1** in 100 mM LiClO$_4$ and acetate buffer (pH 3.3). Two reversible redox waves (at −50 mV and −590 mV) correspond to the Fe(III/II) redox couple, and to 1-e$^-$ reduction of W(VI) in [α-PW$_{11}$O$_{39}$Fe]$^{4-}$ on the surface of the hematite cores. The second (more negative) redox couple of the W(VI) atoms is not well resolved due to overlap with a large cathodic current at −700 mV (possibly due to H$_2$ formation, and/or to reduction of Fe(III) in the hematite cores). (See Supplementary Figure 15b for comparison to [(α-PW$_{11}$O$_{39}$Fe$^{III}$)$_2$O]$^{10-}$ (**2**.)) **d** The surface zeta potentials (ζ, in units of mV) of colloidal α-Fe$_2$O$_3$ (suspension; gray squares) are positive at low pH values and negative at pH values above the isoelectric point near pH 5.5, as indicated by the dotted lines. By contrast, **1** (red diamonds) is soluble and negatively charged from pH 2.5 to 8. (Atomic connectivity of the cluster-anion ligands to the hematite surface is discussed below.)

aggregation differs dramatically from the properties of electrostatically stabilized colloids. Behaving more like a molecular macroanion, the hydrated Na$^+$ salt of **1** can be stored indefinitely, and then readily dissolved in water. These findings provided the first indication that, as shown in Fig. 1a, heteropolytungstate cluster anions are covalently bound to the hematite cores.

**Cluster-anion ligands coordinated to hematite cores of 1.** Energy-dispersive X-ray spectroscopic (EDX) analysis of **1** (Fig. 2a and Supplementary Figure 12) revealed an atomic composition of 38 ± 5% W and 62 ± 5% Fe. Because the cores are crystalline α-Fe$_2$O$_3$, the tungsten atoms must reside at the hematite surface. The tungsten-based surface structures were characterized by electrospray ionization mass spectroscopy (ESI-MS). Mild HCl etching of **1** at pH 2 (6 h at 70 °C), gave a cloudy orange solution (Supplementary Figure 13). After removing an insoluble iron oxide by centrifugation, a pale-yellow powder precipitated upon addition of tetra-*n*-butylammonium (TBA) salt. ESI-MS spectra of this solid revealed two dominant sets of signals, centered at *m/z* = 3981.56 and 2232.57 amu, with isotope-distribution patterns matching the +1 ion, [(TBA)$_5${P-W$_{11}$O$_{39}$Fe(H$_2$O)$_2$}]$^+$ (Fig. 2b) and the 2+ ion, [(TBA)$_7${P-W$_{11}$O$_{39}$Fe(OH)(H$_2$O)}]$^{2+}$, respectively (Supplementary Figure 14). Both ions are consistent with HCl cleavage of [α-PW$_{11}$O$_{39}$Fe] moieties from the surfaces of the hematite cores.

Differential pulse voltammetry (DPV)[25] performed after adding LiClO$_4$ (0.1 M) to aqueous solutions of **1** (Fig. 2c),

provided further support for the presence of [α-PW$_{11}$O$_{39}$Fe] anions at the hematite surface. While conceptually analogous to electrochemical studies of ferrocene-covered SiO$_2$ nanoparticles[33], to our knowledge, the observation of reversible redox chemistries of ligands bound to metal-oxide NCs in solution has few precedents in the literature[25]. It was made possible here by the abundance of cluster anions on the hematite surface, in combination with the water solubility imparted by their negative charges and numerous alkali-metal counter-cations. After adding LiClO$_4$ (0.1 M), differential pulse voltammetry (DPV) revealed reversible redox processes consistent with the presence of the cluster anions identified by ESI-MS (DPV data for the independently prepared molecular cluster anions are provided in Supplementary Figure 15b).

Surface coverage by the POM anions, based on a reasonable footprint of 1.9 nm$^2$, places 15 POMs on the surface of an (idealized) spherical 3-nm-diameter hematite core (Supplementary Table 1). Such a structure would contain 165 W atoms and 279 Fe atoms, giving a relative atomic composition of 37% W and 63% Fe, nicely matching the %-atom values observed by EDX (Fig. 2a). The W to Fe ratio also gives ca. 18–20 Fe atoms per POM ligand.

Not only is **1** remarkably stable to aggregation, it is soluble at pH values of 2.5–8. Over this wide pH range, its largely negative zeta potential values (ζ, −35 to −40 mV) remain pH-invariant (Fig. 2d). By contrast, colloidal α-Fe$_2$O$_3$ precipitates at its isoelectric point (pH 5.5), at which ζ = 0 mV[34]. These findings

represent three independent lines of evidence for strong coordination of the POM ligands to the $\alpha$-Fe$_2$O$_3$ cores.

Given these inert linkages, metathesis of Na$^+$ counter-cations (of the POMs) by $n$-R$_4$N$^+$ (R = hexyl or octyl) was used to render **1** organic-solvent soluble. This cation-exchange procedure—typical of POM salts—gave optically transparent MeCN or MeOH solutions with no loss of POM ligands (Supplementary Figure 16), further evidence for their covalent attachment to the hematite cores.

It is extraordinarily difficult to precisely determine the atomic connectivities of molecules bound to colloidal NCs. However, the presence of a $\mu_2$-oxo linkage between Fe(III) ions in the known oxo-bridged dimer, $[(\alpha$-PW$_{11}$O$_{39}$Fe)$_2$–$\mu_2$-O]$^{10-}$ (**2**)[35], suggested that a similar linkage might bind oxo-donating $[\alpha$-PW$_{11}$O$_{39}$Fe$^{III}$]$^{4-}$–O$^-$ ligands to Fe(III) atoms at the $\alpha$-Fe$_2$O$_3$ surface. X-ray photoelectron spectroscopy (XPS) data for **1** revealed a 2.1 eV difference between W4f$_{7/2}$ and W4f$_{5/2}$ binding energies (Supplementary Figure 17). Notably, these W4f peaks are comparable with those of **2** (Supplementary Figures 18 and 19), consistent with the presence of Keggin-anion derived ligands connected to the surface of **1** via $\mu_2$-oxo linkages.

This was investigated in more detail by FTIR spectroscopy (Fig. 3). The IR-allowed vibrational modes of the central PO$_4$ moieties within Keggin-derived structures are highly sensitive to changes in symmetry of the cluster anion. Removal of a single W (VI) ion from the plenary-Keggin anion, $[\alpha$-PW$_{12}$O$_{40}$]$^{3-}$, reduces the local symmetry of the central PO$_4$ moiety from $T_d$ to $C_{3v}$. As a result, rather than a single IR-active mode at 1080 cm$^{-1}$, two IR-active PO$_4$ modes are observed for mono-lacunary $[\alpha$-PW$_{11}$O$_{39}$]$^{7-}$. And, after complexing a transition-metal cation, the difference in energy between the two IR-active modes depends on cation size, and the degree to which it fits within the lacunary anion's pentacoordinate binding site[36]. As the ion is pulled farther out from the binding pocket, the local symmetry around the central PO$_4$ moiety deviates from $Td$, leading to a larger difference in energy between the two IR-active PO$_4$ bands. Hence, splitting of the PO$_4$ mode can be used to identify the ligand environments of substituted transition-metal cations.

As such, the FTIR spectrum of **1** (Fig. 3a; after baseline correction as shown in Supplementary Figure 20a), was compared with those

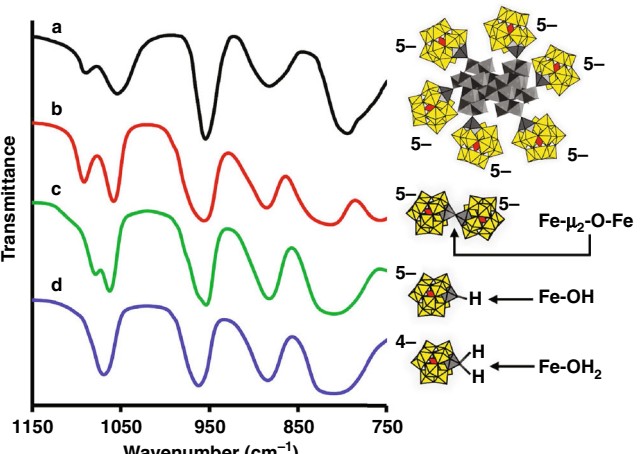

**Fig. 3** FTIR spectral evidence for Fe–$\mu_2$-O–Fe linkages to $\alpha$-[PW$_{11}$O$_{39}$Fe]–O$^-$ ligands in **1**. **a** and **b** FTIR spectra, respectively, of **1** (baseline corrected—see Supplementary Figure 20a) and the $\mu_2$-oxo-bridged dimer, [($\alpha$-PW$_{11}$O$_{39}$Fe)$_2$–$\mu_2$-O]$^{10-}$ (**2**). (The lowest-energy band in **a** is distorted by overlap with bands from $\alpha$-Fe$_2$O$_3$). The IR-active PO$_4$ modes in **a** and **b** match one another, but differ substantially from those of monomeric [$\alpha$-PW$_{11}$O$_{39}$Fe$^{III}$L]$^{n-}$, L = hydroxo or aquo (**c** and **d**, respectively)

of the $\mu$-oxo-bridged dimer, $[(\alpha$-PW$_{11}$O$_{39}$Fe$^{III}$)$_2$–$\mu_2$-O]$^{10-}$ (**2**; Fig. 3b), and of monomeric POMs with aqua and hydroxo ligands, Fe$^{III}$(OH$_2$) and Fe$^{III}$(OH) (Fig. 3, c and d).

The precise match between the PO$_4$ bands[36] of the POM ligands on **1** (1090 and 1051 cm$^{-1}$; Fig. 3a) with the corresponding PO$_4$ bands of **2** (1090 and 1050 cm$^{-1}$; Fig. 3b), is consistent with $\mu_2$-oxo linkages between the POM ligands and the hematite cores. Combined with the EDX data in Fig. 2a, this reasonable atomic connectivity suggests structures of the form, $\{[\alpha$-PW$_{11}$O$_{39}$Fe]-$\mu_2$-O$\}_{\sim 15}$-($\alpha$-Fe$_2$O$_3$)$_{\sim 150}$, in which ca. 15 POMs are bound to cores comprised of ca. 150 Fe$_2$O$_3$ units.

This covalent attachment of POM cluster anions is fundamentally distinct from the use of POMs to electrostatically stabilize colloidal metal-oxide NCs. Notably, in a series of seminal papers, Murray[37], Milleron and Helms[38], and Talapin[39] used NOBF$_4$, Me$_3$OBF$_4$, and Ph$_3$CBF$_4$, respectively, to replace organic protecting ligands on metal-oxide NCs by electrostatically associated BF$_4^-$ anions in polar organic solvents[40]. Milleron[41,42] and Talapin[39] then replaced the weakly bound BF$_4^-$ anions by hexaniobate and other POMs, respectively, giving clear solutions of NCs electrostatically stabilized by the POM anions. Having removed the more tightly bound organic ligands, Milleron used hexaniobate-stabilized Sn-doped In$_2$O$_3$ (ITO) NCs as building blocks for tunable nanocrystal-in-glass composites[41,42], while Talapin showed that films prepared by depositing POM-stabilized Fe$_2$O$_3$ on ITO electrodes were more effective electrocatalysts for water oxidation than analogous ones prepared from organic-ligand protected NCs[39].

In the present work, the covalent attachment of POM ligands to the hematite cores of **1** gives substitutionally inert structures that occupy a unique position at the interface between molecular iron-oxide clusters and electrostatically stabilized colloidal iron-oxide NCs. We now show that, when used as a soluble photocatalyst for visible-light driven water oxidation, **1** is inherently stable under turnover conditions in water.

**Catalytic water oxidation.** The activity of **1** as a visible-light water-oxidation catalyst was explored by irradiating pH-8 solutions of **1** with visible light (150 W Xe lamp, cutoff $\lambda \geq 420$ nm, Supplementary Figures 21 and 22). Four oxidants were evaluated: Ce(IV), Ag$^+$, S$_2$O$_8^{2-}$, and periodate (I$^{VII}$O$_4^-$). Ce(IV) was not suitable as it is only stable at pH values below 1, at which hematite itself dissolves. Although Ag$^+$ is soluble at neutral pH values, these large cations form strong ion pairs with the negatively charged POM ligands, leading to precipitation at desired Ag$^+$ concentrations. By contrast, **1** is soluble in 20 mM solutions of the anionic oxidants. For S$_2$O$_8^{2-}$ and IO$_4^-$, 1025 and 6330 $\mu$mol O$_2$ per gram of $\alpha$-Fe$_2$O$_3$ in **1**, respectively, were produced in 8-h reactions (Table 1, entries 1 and 2; Supplementary Figure 23).

Although written as IO$_4^-$, a recent report shows[43] that its dominant form in water is actually orthoperiodate, H$_5$IO$_6$, which behaves as a polyprotic acid, with pKa$_1$, pKa$_2$, and pKa$_3$ values of ca. 1, 7.5, and 11, respectively. For simplicity, entries in Table 1 refer to added periodate (IO$_4^-$).

The greater reactivity of IO$_4^-$ relative to S$_2$O$_8^{2-}$ (Table 1) is not unique to visible-light driven reactions of **1**. Notably, IO$_4^-$ is also more effective than S$_2$O$_8^{2-}$ in trapping photoexcited electrons from visible-light irradiated WO$_3$[44], Ru(bpy)$_3^{2+}$, and Fe(bpy)$_3^{2+}$[45]. Specifically, visible-light driven electron transfer (ET) from Ru(bpy)$_3^{2+}$ to IO$_4^-$ is twice as fast as the corresponding reduction of S$_2$O$_8^{2-}$, and for Fe(bpy)$_3^{2+}$, visible-light driven ET to IO$_4^-$ is two orders of magnitude faster than ET to S$_2$O$_8^{2-}$[45].

**Support for periodate acting as a two-electron acceptor.** For the reaction in entry 2, two equivalents of IO$_4^-$ are reduced to IO$_3^-$

**Table 1 Visible-light driven water oxidation by 1[a]**

| Entry | Oxidant[b] | Catalyst | Light[c] | Solvent | $O_2$ ($\mu mol\ g^{-1}$)[d] |
|---|---|---|---|---|---|
| 1 | $S_2O_8^{2-}$ | **1** | Visible | $H_2O$ | 1025 |
| 2 | $IO_4^-$ | **1** | Visible | $H_2O$ | 6300 |
| 3 | None | **1** | Visible | $H_2O$ | 0 |
| 4 | $IO_4^-$ | None | Visible | $H_2O$ | 0 |
| 5 | $IO_4^-$ | **1** | Dark | $H_2O$ | 0 |
| 6 | $IO_4^-$ | $\{[\alpha\text{-PW}_{11}O_{39}Fe]_2O\}^{10-}$ (**2**)[e] | Dark | $H_2O$ | 60 ± 20 |
| 7 | $IO_4^-$ | $\{[\alpha\text{-PW}_{11}O_{39}Fe]_2O\}^{10-}$ (**2**)[e] | Visible | $H_2O$ | 60 ± 20 |
| 8 | $IO_4^-$ | $Fe^{3+}$ [f] | Visible | $H_2O$ | 0 |
| 9 | $IO_4^-$ | **1**[g] | Visible | Dry MeCN | 0 |
| 10 | $IO_4^-$ | **1**[g] | Visible | 1:1 $H_2O$:MeCN | 1640 |

[a]All reactions were carried out at pH 8 for 8 h (40 ± 1 °C)
[b]Oxidant concentrations were 20 mM
[c]Light source was a 150 W Xe lamp with a $\lambda \geq 420$-nm cutoff filter
[d]Values reported are per gram of the $\alpha$-Fe$_2$O$_3$ cores, or of the catalysts listed in column three
[e]These two control experiments (entries 6 and 7) rule out oxygen evolution by reaction of periodate with the Fe(III) atoms complexed within the hematite-bound POM ligands. They were carried out at pH 5 to ensure integrity of the molecular dimer, **2**. In 8 h at pH 5, **1** gave 3200 µmol O$_2$ g$^{-1}$
[f]After air oxidation of FeSO$_4$ at pH 5 and 8
[g]Carried out using an organic-solvent soluble form of **1**, and R$_4$N$^+$IO$_4^-$ (R = n-butyl)

for each equivalent of evolved $O_2$ (Supplementary Figure 24), consistent with $IO_4^-$ acting as a 2-e$^-$ oxidant. Moreover, all three components ($IO_4^-$, **1**, and visible light) are required for $O_2$ formation (entries 3–5), and the quantum yield (3.9%) is typical of water oxidation by hematite nanocrystals (Supplementary Table 2). At the same time, when using $IO_4^{-11}$, decomposition or oxo-transfer reactions[46], that could generate $O_2$ without removing four electrons from water, must be ruled out[47,48]. This required numerous control experiments (see Supplementary Figure 25 and Table 3), necessitated by the fact that the oxide ligands of periodate equilibrate rapidly with water, precluding the use of $^{18}O$-labeling to definitively prove that the oxygen atoms in evolved $O_2$ originate from $H_2O$[49]. The results of the control experiments performed are fully consistent with the clean two-electron reduction of $IO_4^-$ to $IO_3^-$.

In brief, photochemical decomposition of $IO_4^-$ (capable of generating $O_2$) occurs only under UV irradiation[50]. When irradiated with visible light ($\lambda \geq 420$ nm), no decomposition of $IO_4^-$ was observed, and no $O_2$ was detected (Table 1; entry 4). Secondly, oxo-transfer mechanisms (noted above) invariably involve thermal (dark) reactions of molecular Fe, Ru, and Ir complexes[46–48,51]. By contrast, dark reactions of $IO_4^-$ with **1** gave no $O_2$ (entry 5), while the dark reaction of $IO_4^-$ with {[$\alpha$-PW$_{11}$O$_{39}$Fe]$_2$O}$^{10-}$ (**2**) gave only traces of $O_2$ (entry 6). The same reaction in visible light (entry 7) also gave traces of $O_2$, probably due to the dark reaction. These findings definitively rule out catalysis by the [$\alpha$-PW$_{11}$O$_{39}$Fe$^{III}$]$^{4-}$–O$^-$ ligands bound via oxo linkages to the $\alpha$-Fe$_2$O$_3$ surface, including via oxo transfer from periodate to the POM-complexed Fe(III) atoms. In addition, no $O_2$ is produced in dark reactions of $IO_4^-$ with $\gamma$-Fe(O)OH[52] or $\alpha$-Fe$_2$O$_3$[28], and no $O_2$ was detected when $IO_4^-$ was reacted under visible light with partially hydrolyzed $Fe^{3+}$ (entry 8).

To confirm that ET to $IO_4^-$ does not generate $O_2$, periodate was stoichiometrically reduced to $IO_3^-$ by two equivalents of the 1-e$^-$ donor, [$\alpha$-AlV$^{IV}$W$_{11}$O$_{40}$]$^{7-}$[53]. No $O_2$ was detected (Supplementary Figure 26a). To model the trapping of photoexcited electrons from **1**, [Ru(bpy)$_3$]$^{2+}$ was reacted with $IO_4^-$. While no reaction occurred in the dark, visible-light ($\lambda \geq 420$ nm) promoted the 2-e$^-$ reduction of $IO_4^-$ to $IO_3^-$. Again, no $O_2$ was detected (Supplementary Figure 26b). Finally, direct evidence that water is required for $O_2$ formation was obtained by reacting organic-solvent-soluble forms of **1** and $IO_4^-$ in dry and wet MeCN (entries 9 and 10; see Supplementary Figure 27)[51]. Although less definitive than isotope labeling, the results show that $H_2O$ is necessary for $O_2$ formation, consistent with $H_2O$ as the source of $O_2$ in pure water.

**Rate optimization and stability under turnover conditions**. Unlike most polyoxometalate-based water-oxidation catalysts, **1** is stable in water over a wide range of pH values, from 2.5 to 8. And, in contrast to colloidal hematite that precipitates from solution at its isoelectric point (i.e., at pH 5.5), **1** is soluble over this entire pH range (see Fig. 2d). Hence, unlike most purely molecular or traditional colloidal catalysts, the activity of **1** can be investigated over a wide range of pH values. The results (Fig. 4a) reveal an approximate doubling of rate from pH values of 5 to 8. At pH 8, the rate of $O_2$ formation is 800 ± 50 µmol g$^{-1}$ h$^{-1}$ (at $\lambda \geq 420$ nm using a 150 W Xe lamp), somewhat exceeding the fastest reported rates for visible-light driven water oxidation by colloidal $\alpha$-Fe$_2$O$_3$ (see Supplementary Table 4 for reported values and related light sources).

The change in pH from 5 to 8 spans the isoelectric point of hematite (near pH 5.5; Fig. 2d). As such, the increase in rate may correlate with deprotonation of water molecules bound to the hematite surface, to give more reactive (negatively charged) hydroxide ligands, in combination with the more favorable, pH-dependent Gibbs free energy for water oxidation itself. For periodate to trap photoexcited electrons, it must diffuse to within a close proximity to the $\alpha$-Fe$_2$O$_3$ surface. The cluster-anion ligated surface of **1**, including [H$_3$I$^{VII}$O$_6$]$^{2-}$ (orthoperiodate—the dominant periodate species present at pH 8—drawn to scale), and Na$^+$ counter-cations (without their hydration shells), is illustrated in Fig. 4b, with iron atoms at the hematite surface shown with the terminal hydroxide ligands that dominate at pH 8. Orthoperiodate is smaller than hydrated Na$^+$ cations;[54] the radius of six coordinate I(VII) is 0.67 Å, compared to 1.16 Å for Na(I) in the aquo complexes, [Na(H$_2$O)$_6$]$^+$, likely located between the negatively charged POM ligands. As such, [H$_3$I$^{VII}$O$_6$]$^{2-}$ anions, and/or hydrogen-bonded ion pairs such as [(H$_2$O)$_5$NaOH$_2$–O$_2$I(O)(OH)$_3$]$^-$, should easily approach Fe$^{III}$–OH moieties at the hematite surface.

The rate of visible-light driven water oxidation by **1** is enhanced by the relatively small size of the hematite cores, which is critically important[28] due to the short, 2–4 nm, hole-diffusion length[55] of $\alpha$-Fe$_2$O$_3$. This leads to a quantum yield of 3.9%, closely matching values reported for optimized reactions of similarly sized colloidal hematite[28,56].

A preliminary schematic of reasonable mechanistic steps is provided in Fig. 4c. Trapping of an excited electron by [H$_3$I$^{VII}$O$_6$]$^{2-}$ (the dominant form of periodate at pH 8)[43] would result in oxidation of a surface Fe$^{III}$–OH moiety to Fe$^{IV}$=O (**A** and **B**, respectively, in Fig. 4c), a species recently

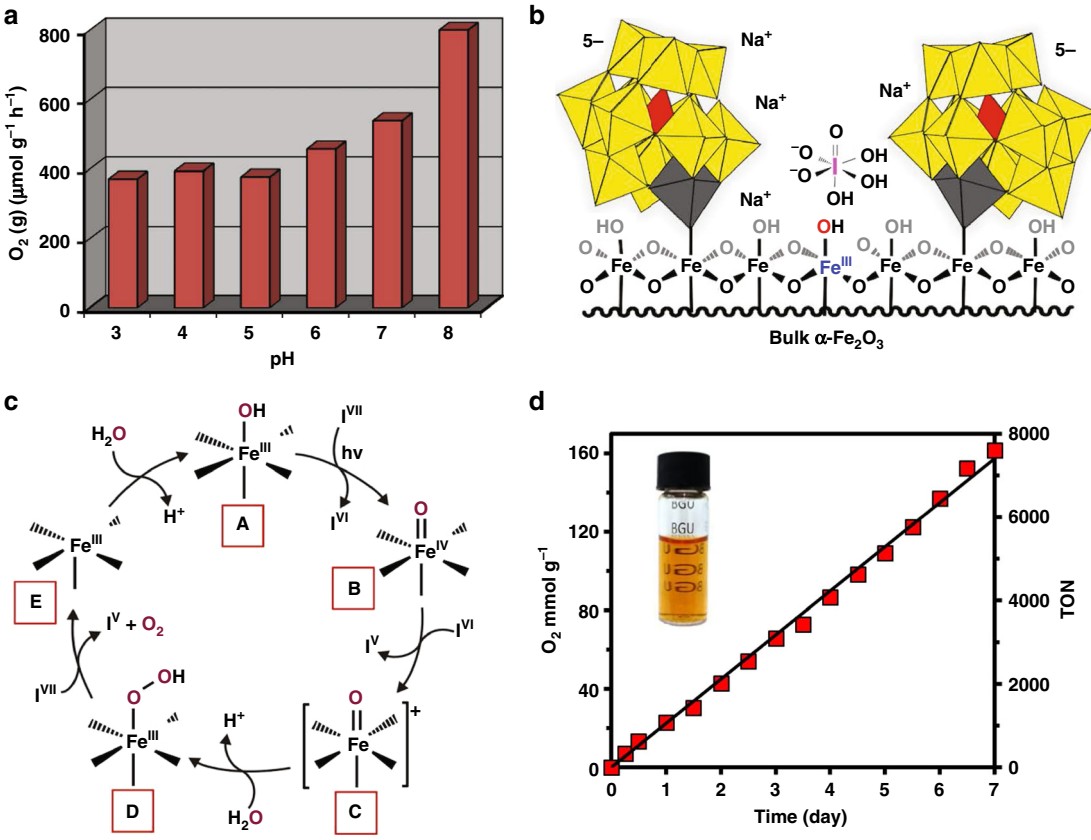

**Fig. 4** Visible-light driven water oxidation by **1**. **a** Rate of O₂ formation (per gram of α-Fe₂O₃ in **1**, per hour), for 2.4 µM **1** and 20 mM NaIO₄, as a function of pH. **b** Scale model showing the relative sizes and plausible locations of cluster-anion ligands, [α-PW₁₁O₃₉Fe]–O⁻, Na⁺ counter-cations, and orthoperiodate ([H₃IᵛᴵᴵO₆]²⁻) at a surface of the α-Fe₂O₃ core of **1**. The Fe(III) atoms at the hematite surface are six coordinate, and at pH 8, are terminated by hydroxide ligands. **c** A plausible catalytic cycle for photochemical water oxidation, highlighting proposed elementary steps, with the active site comprised of a single Feᴵᴵᴵ atom at the surface of the Fe₂O₃ core (highlighted by blue text in panel b). See text for details. **d** Dioxygen produced by **1** (5.4 µM) as a function of time during 7 days of reaction at pH 8. Oxygen production is plotted in mmol O₂ per g of α-Fe₂O₃ in **1** (as routinely done for colloidal metal-oxide catalysts), and the turnover number (TON) is defined as mol O₂ per mol **1**, as is typical for molecular catalysis. The initial concentration of NaIO₄ was 20 mM. During the course of the 1-week reaction, the solution was periodically charged with additional NaIO₄, and separated from accumulated iodate (IO₃⁻) (see the Methods section for details). The starting pH value and those after 7 days, were 8.0 and 8.2, respectively; the buffering was provided by orthoperiodate, for which pKa₂ and pKa₃ are ca. 7.5 and 11, respectively[44]

confirmed by operando infra-red spectroscopy[27] to be an intermediate in photoelectrochemical water oxidation on hematite films[57,58]. The subsequent one-electron oxidation of Feᴵⱽ=O by the reactive I(VI) radical (in a dark reaction) would give a species abbreviated as [Fe=O]⁺ (**C**), whose precise electronic structure is unknown[27,59]. Computational results argue that distances between adjacent Fe atoms at the hematite surface are too large to give stable peroxide-bridged di-iron intermediates, Feᴵᴵᴵ-OO-Feᴵᴵᴵ[60]. At the same time, the first and third-order dependence of hole formation reported by Durrant[61], and the first- and second-order kinetics and operando observations of electrochemical water oxidation by Chen and Song[62], support O–O formation both at a single Fe atom, and at larger hole densities and/or highly basic pH values, via coupling between adjacent surface-trapped holes. For solutions of **1** at pH 8, O–O formation is more likely to occur at a single Fe atom, giving the iron(III)-hydroperoxide intermediate, Feᴵᴵᴵ-OOH (**D**). Dioxygen is then generated by two-electron oxidation of Fe(III)-bound hydroperoxide ligand, resulting in **E**, which reacts rapidly with water to give **A**.

Throughout four consecutive 8-h reactions (a total of approximately 1400 turnovers per equivalent of **1** over 32 h), the turnover frequency (TOF) remained constant (44 ± 1 h⁻¹) and

the solutions remained optically transparent. After each 8-h reaction, the catalyst was quantitatively isolated by salting out with NaCl (2 M) and centrifugation, followed by re-dissolution in fresh water (Supplementary Figure 28a). Notably, this had no effect on its activity. The same remarkable retention of activity was observed in a single 7-day (1-week) reaction (7600 turnovers; Fig. 4d). No decrease in activity was observed. The TOF remained constant at 44 ± 1 h⁻¹, the solution remained optically transparent (inset to Fig. 4d), and no aggregation was observed by DLS (Supplementary Figure 29).

The TON of 7600 is much larger than that reported for colloidal hematite. For example, under identical conditions, the initial rate of O₂ formation by 5-nm colloidal α-Fe₂O₃ (no POM ligands)[56] was ca. 320 µmol g⁻¹ h⁻¹, but decreased dramatically within a few hours due to aggregation processes typical of colloidal metal oxides in water (Supplementary Figure 28b). Moreover, the present TON is more than seven times that reported for the most stable, water-soluble Fe-based catalysts[11] which, although much faster than **1** (with respect to TOF), become inactive within a few hours under turnover conditions in water. And, while acknowledging that **1** contains ca. 300 Fe atoms, many more than typically found in traditional molecular catalysts, its relatively small (ca. 20 Å) α-Fe₂O₃ center is

nevertheless large enough to retain the photochemical properties of bulk hematite. It is from this perspective that **1** can be viewed as a soluble complex of a reactive metal-oxide core.

## Discussion

Oxidatively inert heteropolytungstate cluster anions serve as covalently coordinated oxo-donor ligands for 3-nm hematite (α-$Fe_2O_3$) cores, giving anionic complexes soluble in water over a wide range of pH values (from 2.5 to 8), and that catalyze visible-light driven water oxidation with no need for added photosensitizers. Like molecular macroanions, **1** is highly resistant to aggregation processes that typically lead to the precipitation of electrostatically stabilized hematite and other colloidal metal oxides. And, formed by reaction with entirely inorganic tungsten-oxide-based ligands at 220 °C in water, **1** is inherently (thermodynamically) stable to the oxidative and hydrolytic processes that can limit the active lifetimes of molecular water-oxidation catalysts. As such, **1** can continuously catalyze water oxidation for 7 days (1 week) with no detectable decrease in activity. Moreover, the method used to prepare **1** is not limited to iron oxide, but can be modified according to the pH-controlled aqueous speciation chemistries of numerous other transition-metal ions. As such, the covalent coordination of oxidatively inert polyoxometalate ligands to metal-oxide nanocrystal cores represents a conceptually new and general approach to the design of inherently stable water-oxidation catalysts.

## Methods

**Preparation of 1.** Iron(II) sulfate ($FeSO_4 \cdot 7H_2O$, 16.7 mg, 0.06 mmol) was dissolved into 8 mL of deionized water, and stirred for 1 hour, giving a pale-yellow solution. Freshly prepared 0.2 N KOH (2.2 equiv, 700 μL) was then added dropwise with vigorous stirring, resulting in the formation of an orange-red pH 7.0 suspension of partially amorphous γ-FeO(OH) (s). After 20 h of vigorous stirring at room temperature, 750 μL of 40 mM $Na_7[\alpha\text{-}PW_{11}O_{39}]$ (0.03 mmol) was added dropwise, and 550 μL of deionized water was added to reach a final volume of 10 mL, and stirred for 1 h at room temperature, giving an orange pH 6.7 suspension. The suspension (6 mM Fe(III) and 3 mM $Na_7[\alpha\text{-}PW_{11}O_{39}]$), was then heated for 24 h at 220 °C in a 23-mL Teflon-lined 316 stainless-steel reaction vessel, to give a clear orange-red, pH-6.5 solution of **1**. Details of isolation and purification are provided as Supplementary Materials and Methods.

**Visible-light driven water oxidation.** Photochemical $O_2$ production was carried out in a gas-tight quartz cuvette connected to an upper glass bulb with a headspace volume of 16 mL. Carefully weighed samples of **1** were dissolved in 2.7 mL of water along with sodium periodate ($NaIO_4$; 12.8 mg in 0.1 mL of water) to give a final $NaIO_4$ concentration of 20 mM. The addition of $NaIO_4$ led to a drop in pH from 6.5 to 5, and either 0.2 N KOH or 0.2 N HCl solutions were used to adjust the pH to desired values (see Fig. 4a). The headspace within the cuvette was evacuated and refilled with pure Ar (**g**) four times to remove most of the air, after which the solution was purged with Ar (**g**) for 30 min to remove dissolved oxygen. The solution was then irradiated with a 150 W Xe lamp (USHIO Inc. Japan), after inserting a 420-nm long-pass filter between the light source and the cuvette, which was held at a distance of 12 cm from the light source. The illumination area was 2 $cm^2$ and the average light intensity was 565 mW $cm^{-2}$ at $\lambda = 450$ nm measured using a 1928-C Optical Power Energy Meter (Newport Corp.) equipped with a model 919P-250-35 Thermopile sensor (UV-calibrated silicon detector). The headspace gas (0.5-mL volume portions) was injected every 2 hours into a gas chromatograph (Focus GC, Thermo Scientific) operating at isothermal conditions (40 °C) using a ShinCarbon ST micropacked column (0.53-mm diameter, 2-m length), and equipped with a thermal conductivity detector (TCD) and Ar as carrier gas. Moles of $O_2$ per gram reported for **1** were calculated based on the mass of its α-$Fe_2O_3$ cores, which comprised ca. 50% of the total mass of **1**.

**Water oxidation for 7 days under turnover conditions.** A solution of **1** (5.4 μM) and $NaIO_4$ (20 mM) in 3 mL of water at pH 8 (adjusted using 0.2 N KOH) was degassed as described above and irradiated with visible light ($\lambda > 420$ nm) for 1 day, during which, amounts of $O_2$ in the headspace were quantified at regular intervals (see Fig. 4d). After 1 day, 6.4 mg of solid $NaIO_4$ dissolved in 0.1 mL of water was added to the solution (an additional 10 mM concentration of $IO_4^-$) and the reaction continued for a second day. After day 2, **1** was quantitatively separated from accumulated iodate ($IO_3^-$; ca. 5.5 mM) by making the solution 0.5 M in NaCl, and isolating the salted-out catalyst by centrifugation. (A control experiment later carried out after recharging the supernatant solution with periodate, followed by

irradiation, showed no activity (i.e., no $O_2$ in 8 h). This demonstrated that possibly unidentified soluble components were not responsible for the catalysis.) The pellet of **1** isolated after day 2 was then dissolved in 3 mL of water containing freshly added periodate (20 mM), adjusted to pH 8, degassed, and irradiated for day 3. (Similar separations of **1** from accumulated $IO_3^-$ were repeated after days 4 and 6.) After days 3 and 5, additional 10 mM concentrations of $IO_4^-$ were added. Turnover numbers were calculated based on the moles of $O_2$ produced per mole of **1**.

## Data availability

All data generated or analyzed during this study are included in this published article (and its supplementary information files).

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

## Acknowledgements

I.A.W. thanks the Israel Science Foundation (170/17) for support and B.C. acknowledges the Council of Higher Education Israel for a postdoctoral fellowship.

## Author contributions

I.A.W. directed the research, I.A.W. and B.C. designed the experiments, B.C. carried out the experimental work, M.R., G.G.-O., and E.G. performed electron microscopy, and I.A. W. and B.C. interpreted the data and wrote the paper.

## Additional information

**Competing interests:** The authors declare no competing interests.

