## [Peer Review File · Nature Communications]

Reviewers' Comments:

Reviewer #1:

Remarks to the Author:

This paper describes a hybrid molecule based in the combination of a hematite nanoparticle and tungsten polyoxometallates. The ligation of the POM ligands to the hematite core confers solubility in water and prevents aggregation. Both the hematite and the POM are hydrolytically and oxidatively robust in a wide range of pH. These features represent a very substantial improvement in the physicochemical properties of hematites when subjected to experimental conditions suitable for catalytic water oxidation.

The authors provide solid experimental support for characterizing the new hybrids, and demonstrate their integrity after multiple precipitation, re-dissolution cycles. Notoriously, the authors show that the catalyst remains active during three days, with any sign of decay in the activity, and large TN are obtained. The improved performance of the current hybrids in comparison with colloidal hematite is clearly evidenced.

One of the most interesting aspects in my opinion is the existence of covalent bonds between W atoms of the POC and iron centers at the surface of the hematite particle. This aspect makes 1 a conceptually new class of molecular water oxidation catalysts, just at the interphase of homogeneous and heterogeneous systems. The benefits of the metal-oxides in heterogeneous catalysis is nicely translated into the homogeneous phase.

I believe the work is of high interest for the field of water oxidation catalysis and in general for catalysis because it can be envisioned that the same strategy could be applied to other reactions catalyzed by metal-oxides. The conceptual novelty in the nature of the catalyst, translating into improved catalytic properties, the thoughtful characterization and the rigorous analysis of the data makes this work of high scientific quality and interest, in my opinion suitable for Nat. Commun.

Therefore, I recommend publication after the following aspects are clarified.

I understand that the characterization of the hematite surface, specially the proposed Fe-O-W linkage is particularly difficult. Still, I believe that there are some techniques that can help in providing a more definitive conclusion. For example, I miss a W-NMR analysis, which I think should provide evidence for the heteronuclear core, and for the stoichiometry.

The IR characterization needs some clarification. When looking at Fig 3, one may arrive to the conclusion that all the iron sites in the surface of the hematite are oxo-bridged, but this is not the case. Only a small fraction is. Authors need to provide some explanation for this.

The reaction mechanism is quite difficult to understand. A diagram will help very much. In particular, the description of the role of IO₄⁻ in pages 13-14 is confusing. The authors refer "to photochemical decomposition" the productive role of IO₄⁻ in catalysis. It is clear that the reaction is a two e⁻ reduction, but I have not been able to imagine the sequence of elemental reactions leading to reduction of IO₄⁻ and oxidation of water.

I think that the comparison of TN with those of small molecular catalysts is completely fair. To my understanding, the molecular catalysts used for comparison are monometallic, while 1 contains approx. 150 Fe₂O₃ units. If one considers mmols O₂/mmol of Fe then the number becomes modest in comparison with these catalysts. A more balanced discussion is recommended.

Reviewer #2:

Remarks to the Author:

Prof. Weinstock and coworkers established an interesting strategy for increasing the stability of Fe₂O₃ nanoparticles by covalently decorating them with POMs, and obtained composites exhibited high efficiency and durability in water oxidation. The results have been well-proved and the manuscript is also well-documented. While, at the same time, there are still several severe concerns over the

conceptual novelty. Very recently, the functionalization of the nanocrystal (NC) surface with inorganic oxo ligands including POMs have been reported, see ACS Nano, 2014, 8 (9), 9388–9402; especially, water oxidation has already been conducted using Fe₂O₃ nanocrystals decorated with [P₂Mo₁₈O₆₂]⁶⁻ ligands. Moreover, similar trend of stability enhancement after superficial modification was also revealed. Thus, it is quite necessary to demonstrate what is conceptually new in this paper. Major revision and resubmission are needed for further consideration.

Reviewer #3:

Remarks to the Author:

In this manuscript, the authors described a method for covalently attaching polyoxotungstate to hematite cores, giving stable anionic structures (complex 1) that are highly resistant to aggregation and stable to oxidation and hydrolysis (within a certain pH range). Complex 1 is shown to be a highly stable catalyst for visible-light induced water oxidation to O₂ using periodate as the oxidant, with turnovers of over 3000 after irradiation for 3 days. The use of the oxidation resistant polyoxotungstate to bind to hematite so to prevent aggregation is a novel approach to the design of stable water oxidation catalysts. This work should be of interest to researchers in the field of water oxidation, it could provide a new research direction to the general area of photocatalysis.

This paper is expertly written, the catalyst has been rigorously characterized by various techniques, including electron microscopy, mass spectrometry and infrared spectroscopy. The experimental part is clearly written. Extensive experimental data are also provided in Supporting Information.

In the opinion of this reviewer, this paper is potentially suitable for publication in Nature Communications. However, there are some limitations/drawbacks/concerns for this system:

1. The authors seemed to claim that the functions of the Fe(III) substituted polyoxotungstate anions are to solubilize the hematite and prevent aggregation. However, since a number of molecular iron complexes are known to be efficient water oxidation catalysts, it is possible that in this system part of the oxygen evolved may arise from catalysis by Fe(III)POM.
2. The main drawback of this system is that efficient catalysis can also be achieved using periodate (persulfate is much less efficient) as the sacrificial oxidant. There are in fact, at least a couple of examples in the literature (Ref 11 and Chem. Commun. 2015, 51, 12189) of molecular iron complexes that readily catalyze thermal water oxidation by periodate, without the need of light and Fe₂O₃ semiconductor. The turnover numbers for the molecular catalysts are around 1000, which are just 3 times lower than the TON of 3200 for the much more complicated system reported in this paper. The authors claimed that their system is very stable and there is no decrease in catalytic activity after 3200 turnovers and 3 days of irradiation. Perhaps the authors can provide more convincing evidence on the stability of their catalysts, such as by doing catalysis for a longer period of time.

Nature Communications
The Macmillan Building
4 Crinan Street
London N1 9XW, UK

Sept. 25, 2018

Dear Dr. Chen,

Attached please find our revised manuscript, “**Design of an inherently-stable water oxidation catalyst**”, in which polyometalate ligands are covalently attached to small hematite nanocrystals, giving a soluble visible-light driven water-oxidation catalyst with unprecedented stability under turnover conditions in water. In our original manuscript, we operated the catalyst for **three days**. In this revised manuscript, we now report that **no decrease in reactivity is observed after continuous operation for a full week**.

We have responded to every suggestion made by the three reviewers, in each specific case by changes or additions to the Text, Methods and/or Supplementary Information. These include entirely new experiments, graphics, and discussion and references, all highlighted in yellow in the revised manuscript.

Reviewer 1 writes, “*One of the most interesting aspects ...is the existence of covalent bonds between W atoms of the POC and iron centers at the surface of the hematite particle. This aspect makes 1 a conceptually new class of molecular water oxidation catalysts, just at the interphase of homogeneous and heterogeneous systems. ... the work is of high interest [to] water oxidation catalysis and in general for catalysis. ... The conceptual novelty in the nature of the catalyst, translating into improved catalytic properties, the thoughtful characterization and the rigorous analysis of the data makes this work of high scientific quality and interest, in my opinion suitable for Nat. Commun.*”. Items identified for revision were clearly spelled out, and the changes made in response have clearly improved the work.

Reviewer 3 writes, “covalently attaching polyoxotungstate to hematite cores ... giv[es] stable anionic structures (1) ... highly resistant to aggregation and stable to oxidation and hydrolysis ... Complex 1 is ... a highly stable catalyst for visible-light induced water oxidation ..., with turnovers of > 3000 after irradiation for 3 days [This] is a novel approach to the design of stable water oxidation catalysts. [It] should be of interest to researchers in the field of water oxidation [as] a new research direction to the general area of photocatalysis.” The reviewer requested a control experiment (**done!**), and “more convincing evidence on the stability of their catalysts, such as by doing catalysis for a longer period of time.” In response, we increased continuous operation from **three days** (3200 turnovers) to a **full week** (7600 turnovers), again **with no decrease in reactivity**. This is truly dramatic, and we warmly thank this reviewer for suggested we do this.

Reviewer 2's writes, it is “quite necessary to demonstrate what is conceptually new in this paper. Major revision and resubmission are needed for further consideration.” No new experiments or data are requested. This reviewer's brief comments focus on an article by Talapin (2014), who extends a general method reported by Milleron (2011 & 2013), to prepare a variety of colloidal metal oxides—including iron oxide—**electrostatically stabilized** in organic solvents by POM anions. The electrostatically stabilized colloidal iron oxide NCs are drop cast onto an electrode and used as a (dark) electrocatalyst for water oxidation. We explain in detail (below) how our **covalent structure** and **solution-state reactivity** are **both conceptually new**. In an objective and scholarly addition to the revised text (190 words; six new references), we clarify what is conceptually new in our work in relation to Talapin's contribution, which is fully acknowledged.

Point-by-point responses to each of the reviewers' comments are included below.

Thank you.

Sincerely,

Ira A. Weinstock

Point-by-point responses to reviewers 1, 2 and 3

Reviewer 1

Reviewer 1 writes, "The conceptual novelty in the nature of the catalyst, translating into improved catalytic properties, the thoughtful characterization and the rigorous analysis of the data makes this work of high scientific quality and interest, in my opinion suitable for Nat. Commun."

Reviewer 1 clearly understands the context, novelty and significance of the work, and we appreciate his/her thoughtful suggestions for revision. Each has been addressed by additions of new experimental data and/or clarifying discussion, all of which appear in the revised manuscript. These are detailed immediately below.

Comment 1. "I understand that the characterization of the hematite surface, especially the proposed Fe-O-W linkage is particularly difficult. Still, I believe that there are some techniques that can help in providing a more definitive conclusion. For example, I miss a W-NMR analysis, which I think should provide evidence for the heteronuclear core, and for the stoichiometry."

Response: We agree that a ^{183}W NMR spectrum of the POM-complexed hematite, $[\text{PW}_{11}\text{O}_{39}\text{Fe}]^{4-}\text{-O-Fe}_2\text{O}_3$ (**1**), would be nice to have. Unfortunately, the concentrations of POM ligands in solutions of **1** (less than mM) are much too small for the acquisition of ^{183}W NMR spectra. Typically, ca. 0.5 M W concentrations are needed to obtain spectra with discernible ^{183}W peaks, and even at his large concentration, overnight acquisition is typically required. In the present case, the larger relaxation times of POMs bound to nanostructures in solution (in comparison to freely diffusing molecular cluster-anions) would make this even more difficult. In principle, solid-state ^{183}W NMR spectra dried samples would give discernible signals, but those are very broad and vary by many ppm values as functions of local hydration environments and distances from counter-cations. Considering the option of using ^{31}P NMR, this is not feasible due to the presence of paramagnetic Fe(III) (d^5) ions present within the lacunary site of the POM ligands. The paramagnetic iron atoms would probably also compromise the acquisition of useful ^{183}W NMR spectra.

Changes made in revision: Nevertheless, we agree with reviewer **1** that additional spectroscopic data would be helpful. And, while still short of providing a definitive picture of atomic connectivity at the hematite surface, we obtained additional high-resolution XPS data for use in comparing **1** with the dimeric (oxo-bridged) molecular cluster-anion, $[(\alpha\text{-PW}_{11}\text{O}_{39}\text{Fe})_2\text{-}\mu_2\text{-O}]^{10-}$ (**2**). These (reproduced below) are included in the revised manuscript as Supplementary Figs. 16 and 17, and referred to in the text of the revised manuscript itself, in the section on atomic connectivity.

Here is a quote of the added text, followed by copies of the new data now provided as Supplementary Figures: "X-ray photoelectron spectroscopy (XPS) data for **1** revealed a 2.1 eV difference between $\text{W}4f_{7/2}$ and $\text{W}4f_{5/2}$ binding energies (Supplementary Figs. 16 and 17). Notably, these $\text{W}4f$ peaks are comparable to those of **2** (consistent with the presence of Keggin-anion derived ligands connected to the surface of **1** via μ_2 -oxo linkages)."

From Supplementary Fig. 16. High-resolution XPS data for $(\text{TEA})_{10}[(\text{PW}_{11}\text{O}_{39}\text{Fe})_2\text{O}]$. Panels (a) – (d) are scans for Fe2p, W4f, P2p and O1s, respectively."

Point-by-point responses to reviewers 1, 2 and 3

From Supplementary Fig. 17. Comparison of W4f peaks for **1** (top; black curve) and for molecular $[(PW_{11}O_{39}Fe)_2O]^{10-}$ (**2**) (bottom; red curve), showing that both W4f scans resemble one other. "

Comment 2. "The IR characterization needs some clarification. When looking at Fig 3, one may arrive to the conclusion that all the iron sites in the surface of the hematite are oxo-bridged, but this is not the case. Only a small fraction is. Authors need to provide some explanation for this."

Response: Looking now at the original manuscript, it is easy to appreciate reviewer 1's request for clarification. Prior to submission to Nature Comm., we removed a detailed discussion of this set of experiments, simply to comply with the journal's size limitations. In light of reviewer 1's response, it is clear that more discussion is needed to fully understand the data presented. That discussion is now included in the revised manuscript. The main point is that, as the reviewer correctly notes, only a small fraction of the surface Fe(III) atoms are linked via oxo bridges to Fe(III) atoms complexed by the POM ligands. The infrared spectrum of the iron-oxide core in **1** give a broad absorbance from ca. 1000 – 600 cm^{-1} , with the %-transmission decreasing over this range, but presenting no distinct features. The much weaker bands due to the POM ligands—15 of which are bound to a similar number of Fe(III) atoms at the surface of the hematite core—also span this same range in cm^{-1} values. Due to the underlying absorbance arising from the hematite core, the *relatively weak POM bands are resolved by baseline correction*. In this regard, I refer the reviewer to the red and green curves in Supplementary Figure 18a. Once the baseline correction has been made, the bands arising from the POM dominate, giving the spectrum in Fig. 3a (text). So, although this spectrum shows only bands arising from the POM ligands, this is not meant to imply that all the Fe(III) ions at the surface of the hematite cores are linked to the relatively fewer POM ligands.

Changes made: We have now added back the detailed (and more complete and clear) discussion of the FTIR study that was removed prior to submission of the original manuscript, and more discussion has been added in a (now) lengthy caption to Supplementary Figure 18 (i.e., concerning the FTIR study). The revised text and the new caption to Supplementary Figure 18, are as follows (also incorporated here—in red text—is discussion of the newly obtained XPS data mentioned above):

New and revised discussion added to the text:

"It is extraordinarily difficult to precisely determine the *atomic connectivities* of molecules bound to colloidal NCs. However, the presence of a μ_2 -oxo linkage between Fe(III) ions in the known oxo-bridged dimer, $[(\alpha-PW_{11}O_{39}Fe)_2-\mu_2-O]^{10-}$ (**2**),³⁵ suggested that a similar linkage might bind oxo-donating $[\alpha-PW_{11}O_{39}Fe^{III}]^4-O^-$ ligands to Fe(III) atoms at the $\alpha-Fe_2O_3$ surface. X-ray photoelectron spectroscopy (XPS) data for **1** revealed a 2.1 eV difference between $W4f_{7/2}$ and $W4f_{5/2}$ binding energies (Supplementary Figs. 16 and 17). Notably, these W4f peaks are comparable to those of **2**, consistent with the presence of Keggin-anion derived ligands connected to the surface of **1** via μ_2 -oxo linkages.

This was investigated in more detail by FTIR spectroscopy (Fig. 3). The IR-allowed vibrational modes of the central PO_4 moieties within Keggin-derived structures are highly sensitive to changes in symmetry of the

Point-by-point responses to reviewers 1, 2 and 3

cluster-anion. Removal of a single W(VI) ion from the plenary-Keggin anion, $[\alpha\text{-PW}_{12}\text{O}_{40}]^{3-}$, reduces the local symmetry of the central PO_4 moiety from T_d to C_{3v} . As a result, rather than a single IR-active mode at 1080 cm^{-1} , two IR-active PO_4 modes are observed for mono-lacunary $[\alpha\text{-PW}_{11}\text{O}_{39}]^{7-}$. And, after complexing a transition-metal cation, the difference in energy between the two IR-active modes depends on cation size, and the degree to which it fits within the lacunary anion's pentacoordinate binding site.^{36,37} Hence, splitting of the PO_4 mode can be used to identify the ligand environments of substituted transition-metal cations.

As such, the FTIR spectrum of **1** (Fig. 3a; after baseline correction as shown in Supplementary Figure 18a), was compared with those of the μ -oxo-bridged dimer, $[(\alpha\text{-PW}_{11}\text{O}_{39}\text{Fe}^{\text{III}})_2\text{-}\mu_2\text{-O}]^{10-}$ (**2**; Fig. 3b), and of monomeric POMs with aqua and hydroxo ligands, $\text{Fe}^{\text{III}}(\text{OH}_2)$ and $\text{Fe}^{\text{III}}(\text{OH})$ (Fig. 3, c and d).

The precise match between the PO_4 bands³⁶ of the POM ligands on **1** (1090 & 1051 cm^{-1} ; Fig. 3a) with the corresponding PO_4 bands of **2** (1090 & 1050 cm^{-1} ; Fig. 3b), is consistent with μ_2 -oxo linkages between the POM ligands and the hematite cores. Combined with the EDX data in Fig. 2a, this reasonable atomic connectivity suggests structures of the form, $\{[\alpha\text{-PW}_{11}\text{O}_{39}\text{Fe}]\text{-}\mu_2\text{-O}\}_{-15}\text{-}(\alpha\text{-Fe}_2\text{O}_3)_{-150}$, in which ca. 15 POMs are bound to cores comprised of ca. 150 Fe_2O_3 units."

Reference 36 and note 37 (cited in the above text, and added to the revised manuscript) refer to vibrational spectra of transition-metal substituted lacunary-Keggin anions. For convenience, they are reproduced here:

36 Rocchiccioli-Deltcheff, C. & Thouvenot, R. Metal complexes of heteropolyanions $\alpha\text{-XW}_{11}\text{O}_{39}^{n-}$ with $\text{X} = \text{Si}^{\text{IV}}$ or P^{V} and $\text{M} = \text{Mo}^{\text{VI}}$ or W^{VI} : Study of structural modifications of ligand by infrared and raman spectrometry. *J. Chem. Res., Part S (Synop.)*, 46-47, (1977).

37 As the ion is pulled farther out from the binding pocket, the local symmetry around the central PO_4 moiety deviates from T_d , leading to a larger difference in energy between the two IR-active PO_4 bands.

Expanded discussion added to the Supplementary Figure related to the vibrational study: Further discussion (immediately below) has been added to the caption of Supplementary Figure 18 (revised ms).

"Supplementary Fig. 18: Characterization of **1** by vibrational spectroscopy. (a) FTIR spectra of $\alpha\text{-Fe}_2\text{O}_3$ (black curve), **1** unprocessed (red), and after baseline correction (green), and (b), Raman spectra of **1** (green), $\alpha\text{-Fe}_2\text{O}_3$ (black), and $[(\text{PW}_{11}\text{O}_{39}\text{Fe})_2\text{O}]^{10-}$ (blue). The FTIR spectrum of $\alpha\text{-Fe}_2\text{O}_3$ whose (black curve in panel a) was obtained using $\alpha\text{-Fe}_2\text{O}_3$ prepared under the same conditions (concentration, pH, temperature and time) as that used to prepare **1**, but without the presence of POM. The average particle size (see inset TEM image) is much larger than the ca. 1.9 nm radius hematite cores of **1**. In the unprocessed (i.e., non-baseline corrected) FTIR spectrum of **1** (red curve), absorbance from the small iron-oxide NC cores is most pronounced starting from ca. 900 cm^{-1} and increases to a maximum (smaller %-transmittance) at ca. 750 cm^{-1} . These two results (black and red curves) are consistent with FTIR spectra of $\alpha\text{-Fe}_2\text{O}_3$ samples obtained using different preparative methods (and particle sizes), which are reported to give abrupt increases in absorbance (and smooth drops in transmittance) starting in some cases from 850 cm^{-1} (similar to **1**), and from 700 cm^{-1} in others (similar to our independently prepared $\alpha\text{-Fe}_2\text{O}_3$; black curve). The shift in onset of absorbance to larger wavenumber values for **1**, relative to that of pure $\alpha\text{-Fe}_2\text{O}_3$ (i.e., from 700 to ca. 900 cm^{-1}) could be due to the small size of the hematite cores in **1**. The cores of **1** are comprised of ca. 300 Fe atoms, 25% of which (i.e., ca. 75 Fe atoms) are estimated to lie at the particle surface. Of those, ca. 15 are bound to POM ligands. This surface functionalization would be expected to give rise to bands differing in energy from those within bulk $\alpha\text{-Fe}_2\text{O}_3$, and perhaps contributes the shift in onset of the iron-oxide absorbance to 900 cm^{-1} . Baseline correction was used to obtain the green curve, in which the relatively weaker intensity bands arising from the POM ligands are more clearly resolved. This baseline-corrected spectrum is presented in Fig. 3 of the text. (b) Comparison of the Raman spectra of **1** with that of $\alpha\text{-Fe}_2\text{O}_3$ and of the molecular POM reveals the $\text{W}=\text{O}$ stretching band of POM (sharp peak near 945 cm^{-1}) along with standard Raman-allowed stretches of $\alpha\text{-Fe}_2\text{O}_3$ ($200\text{-}600\text{ cm}^{-1}$). The presence of one $\text{W}=\text{O}$ stretching band in **1** instead of two—generally assigned to symmetric and anti-symmetric $\text{W}=\text{O}$ stretches of molecular $[(\text{PW}_{11}\text{O}_{39}\text{Fe})_2\text{O}]^{10-}$ —indicates that the Raman-allowed modes for the POM bound to the hematite surface are different than for the molecular dimer."

Comment 3. "The reaction mechanism is quite difficult to understand. A diagram will help very much. In particular, the description of the role of IO_4^- in pages 13-14 is confusing. The authors refer "to photochemical decomposition" the productive role of IO_4^- in catalysis. It is clear that the reaction is a two e- reduction, but I have not been able to imagine the sequence of elemental reactions leading to reduction of IO_4^- and oxidation of water."

Point-by-point responses to reviewers 1, 2 and 3

Response: The role of IO_4^- in transition-metal catalysed water oxidation reactions has been a recent subject of discussion in the literature. Depending on the reaction conditions and type of catalyst used, IO_4^- can decompose to liberate O_2 , serve as an oxo donor, or act as a two-electron oxidant. At the same time, due to rapid oxide-ligand exchange in water, O-18 labelling can't not be used to clarify the role of IO_4^- in specific reactions. The possible roles of IO_4^- , the need for numerous control experiments, and results of those we carried out, are succinctly summarized in pages 13-14 (including in Table 1). The main point is that all control experiments were fully consistent with IO_4^- trapping photo-excited electrons from the hematite cores of **1**, with subsequent reduction by a second electron giving IO_3^- (iodate). (In this context, "photochemical decomposition" of periodate was one of several possibilities that we ruled out.)

Having confirming that IO_4^- acted as a two-electron oxidant, we moved directly to catalytic results, without elaborating on the mechanism. Notably, however, we have carried out a highly detailed mechanistic study, using a toolbox of solution-state methods typically reserved for molecular catalysts (possible here because of the solubility and stability of **1**). The data and analysis are too lengthy to include in the present communication, and we are preparing a separate manuscript on that topic.

At the same time, we fully agree with the reviewer that it would be helpful to clarify the sequence of elementary reactions involved. We've done this in the revised manuscript, by providing a catalytic cycle (as the requested "diagram"), along with explanatory text. We've tried to strike a balance between highlighting the key elementary steps, while still retaining the option of published the full mechanistic study as a separate article. (A post-doctoral associate in my laboratory worked for over a year on that study, and I trust the reviewer will understand our reluctance to bury it in the Supplementary Information file of the present article.) The newly added diagram (Fig. 4c of the revised manuscript) and added text, are reproduced immediately below. (The previous graphic in Fig. 4c, concerning three consecutive 8-hour reactions, has been moved to the Supplementary information.

Below are the newly added scheme and explanatory text, followed by the newly added references:

Fig. 4. Visible-light driven water-oxidation by **1.** c) A plausible catalytic cycle for photochemical water oxidation, highlighting proposed elementary steps, with the active site comprised of a single Fe^{III} atom at the surface of the Fe_2O_3 core (highlighted by blue text in panel b). See text for details

"A preliminary schematic of reasonable mechanistic steps is provided in Fig. 4c. Trapping of an excited electron by $[\text{H}_3\text{I}^{\text{VII}}\text{O}_6]^{2-}$ (the dominant form of periodate at pH 8)⁴³ would result in oxidation of a surface $\text{Fe}^{\text{III}}\text{-OH}$ moiety to $\text{Fe}^{\text{IV}}\text{=O}$ (A and B, respectively, in Fig. 4c), a species recently confirmed by "operando" infra-red spectroscopy²⁷ to be an intermediate in photoelectrochemical water oxidation on hematite films.^{57,58} The subsequent one-electron oxidation of $\text{Fe}^{\text{IV}}\text{=O}$ by the reactive I(VI) radical (in a dark reaction) would give a species abbreviated as " $[\text{Fe}=\text{O}]^+$ ", whose precise electronic structure is unknown.^{27,59} Computational results argue that distances between adjacent Fe atoms at the hematite surface are too large to

Point-by-point responses to reviewers 1, 2 and 3

give stable peroxide-bridged di-iron intermediates, $\text{Fe}^{\text{III}}\text{-OO-Fe}^{\text{III}}$.⁶⁰ At the same time, the first and third-order dependence of hole formation reported by Durrant,⁶¹ and the first and second-order kinetics and operando observations of electrochemical water oxidation by Chen and Song,⁶² support O-O formation both at a single Fe atom, and at larger hole densities and/or highly basic pH values, via coupling between adjacent surface-trapped holes. For solutions of **1** at pH 8, O-O formation is more likely to occur at a single Fe atom, giving the iron(III)-hydroperoxide intermediate, $\text{Fe}^{\text{III}}\text{-OOH}$ (**D**). Dioxygen is then generated by two-electron oxidation of Fe(III)-bound hydroperoxide ligand, resulting in **E**, which reacts rapidly with water to give **A**."

New references cited the above text:

- 58 Klahr, B. & Hamann, T. Water oxidation on hematite photoelectrodes: Insight into the nature of surface states through in situ spectroelectrochemistry. *J. Phys. Chem. C* **118**, 10393-10399, (2014).
- 59 Hellman, A. & Pala, R. G. S. First-principles study of photoinduced water-splitting on Fe_2O_3 . *J. Phys. Chem. C* **115**, 12901-12907, (2011).
- 60 Bernasconi, L., Kazaryan, A., Belanzoni, P. & Baerends, E. J. Catalytic oxidation of water with high-spin iron(IV)-oxo species: Role of the water solvent. *ACS Catal.* **7**, 4018-4025, (2017).
- 61 Yatom, N., Neufeld, O. & Caspary Toroker, M. Toward settling the debate on the role of Fe_2O_3 surface states for water splitting. *J. Phys. Chem. C* **119**, 24789-24795, (2015).
- 62 Le Formal, F., Pastor, E., Tilley, S. D., Mesa, C. A., Pendlebury, S. R. *et al.* Rate law analysis of water oxidation on a hematite surface. *J. Am. Chem. Soc.* **2015**, 6629-6637, (2015).
- 63 Zhang, Y., Zhang, H., Liu, A., Chen, C., Song, W. *et al.* Rate-limiting O-O bond formation pathways for water oxidation on hematite photoanode. *J. Am. Chem. Soc.* **140**, 3264-3269, (2018).

Comment 4. "I think that the comparison of TN with those of small molecular catalysts is completely fair. To my understanding, the molecular catalysts used for comparison are monometallic, while **1** contains approx. 150 Fe_2O_3 units. If one considers mmols O_2 /mmol of Fe then the number becomes modest in comparison with these catalysts. A more balanced discussion is recommended."

Response: We appreciate the feedback, and are happy to provide a more balanced discussion, noting the much larger number of Fe atoms in our catalyst than in typical molecular catalysts.

Changes made: Newly added text is highlighted in yellow.

"The TON of nearly 7600 is much larger than that reported for colloidal hematite. For example, under identical conditions, the initial rate of O_2 formation by 5-nm colloidal $\alpha\text{-Fe}_2\text{O}_3$ (no POM ligands)⁵⁰ was ca. $320 \mu\text{mol g}^{-1} \text{h}^{-1}$, but decreased dramatically within a few hours due to aggregation processes typical of colloidal metal oxides in water (Supplementary Fig. 27b). Moreover, the present TON is close to eight times that reported for the most stable, water-soluble Fe-based catalysts¹¹ which, although much faster than **1** (with respect to TOF), become inactive within a few hours under turnover conditions in water. And, while acknowledging that **1** contains ca. 300 Fe atoms, many more than typically found in traditional molecular catalysts, its relatively small (ca. 20 Å) $\alpha\text{-Fe}_2\text{O}_3$ centre is nevertheless large enough to retain the photochemical properties of bulk hematite. It is from this perspective that **1** can be viewed as a soluble complex of a reactive metal-oxide core."

Point-by-point responses to reviewers 1, 2 and 3

Reviewer 2

Comment 1. “Prof. Weinstock and coworkers established an interesting strategy for increasing the stability of Fe₂O₃ nanoparticles by covalently decorating them with POMs, and obtained composites exhibited high efficiency and durability in water oxidation. The results have been well-proved and the manuscript is also well-documented. While, at the same time, there are still several severe concerns over the conceptual novelty. Very recently, the functionalization of the nanocrystal (NC) surface with inorganic oxo ligands including POMs have been reported, see ACS Nano, 2014, 8 (9), 9388–9402; especially, water oxidation has already been conducted using Fe₂O₃ nanocrystals decorated with [P₂Mo₁₈O₆₂]⁶⁻ ligands. Moreover, similar trend of stability enhancement after superficial modification was also revealed. Thus, it is quite necessary to demonstrate what is conceptually new in this paper.”

Response: We appreciate this reviewer's positive comments regarding the quality of our results, described as "well proved" and "well documented", and also thank this reviewer for providing us the opportunity to clarify what is "conceptually new" in the paper. The seminal work reported by Talapin (*ACS Nano*, 2014) is indeed relevant to understanding the conceptual advances in our current contribution. As such, we now summarize Talapin's work (along with related work of Murray and Milleron) in the revised text, and highlight the conceptually new aspects of our work. The text added in revision is provided farther below. Immediately below, however, is a more detailed discussion, too lengthy to place in the manuscript itself.

Our work is conceptually new with respect to both **structure** and **reactivity**. It concerns the design and synthesis of an entirely new type of POM-coordinated iron-oxide core, and its use as a soluble non-labile visible-light driven water oxidation catalyst, with inherent (thermodynamic) stability with respect to oxidative and hydrolytic degradation.

Structure. In the *ACS Nano* article by Talapin (referred to by reviewer 2), the POMs are electrostatically associated with positively charged metal-oxide cores. These are a type of traditional electrostatically stabilized colloidal nanocrystals, obtained by exchanging BF₄⁻ anion by POM anions, and soluble in polar organic solvents. As such, they are inherently labile and susceptible to hydrolysis and / or to exchange with other anions (such as the periodate, IO₄⁻, used as an oxidant in the present work).

The **covalent** attachment of POM ligands to hematite cores in our work constitutes a fundamentally different type of structure, and (importantly) provides for **inherent thermodynamic stability in water**. This includes **stability to oxidation, hydrolysis, aggregation and ligand exchange**. This is why we refer to the new structures as occupying a unique position between molecular macroanions and traditional (electrostatically-stabilized) colloids, such as Talapin's.

Reactivity. As a result of this inherent stability, the POM-complexed structures in the present manuscript can be deployed as soluble visible-light driven water-oxidation catalysts. In the limited reactivity test reported by Talapin, no solution-state catalysis is claimed or demonstrated. Rather, the electrostatically stabilized iron-oxide colloid is drop cast onto a working electrode, after which, catalytic currents are observed at positive potentials. No O₂ is actually measured, and no effort is made to demonstrate the integrity of the solid-state electrocatalyst after its use.

By contrast, **our new type of soluble catalyst is inherently stable by design**—and this is proven by operation for one week (7 days) under turnover conditions in water, during which no decrease in reactivity is observed.

In this regard, we would like to share with reviewer 2 some comments on the general context of our work:

The larger context is the yet-unsolved problem of oxidative and / or hydrolytic degradation of **soluble water oxidation catalysts**. While molecular water-oxidation catalysts feature rapid turnover rates, they are inevitably susceptible to degradation under turnover conditions in water. Metallo-organic catalysts are susceptible to oxidation (of their organic ligands), as well as to hydrolysis, while oxidatively-inert polyoxometalate complexes are susceptible to hydrolysis. The degradation processes lead to the formation of reactive colloidal metal oxides whose roles in investigations of molecular catalysts are an ongoing, and often-times heated topic of discussion in the community.

Point-by-point responses to reviewers 1, 2 and 3

We address this head on by using oxidatively inert polyoxometalates (POMs) as **covalently attached** ligands for water-soluble iron-oxide cores, giving nanostructures that are **thermodynamically stable to both oxidation and hydrolysis**, and **highly resistant to aggregation in water**. We covalently attach the POM ligands to the iron-oxide cores at high temperature (220 °C) in water, to give a new type of nanostructure, conceptually located between molecular macroanions and traditional, electrostatically stabilized colloidal metal oxides. We've published extensively on electrostatically stabilized POM ligand shells on Au nanoparticles (*cf. J. Am. Chem. Soc.*, **2009**, *131*, 17412, and *Nature Nanotechnology*, **2017**, *12*, 170) and in that context, we were honestly surprised we could covalently attach POM ligands to metal-oxide NC surfaces!

Hence, prior to our first publication on this new type of structure (*Angew. Chem. Int. Ed.* **2015**, *54*, 12416; cited in original and revised manuscripts), **we spent one and a half years proving to ourselves that the**

POM ligands were in fact covalently attached, in that case, to anatase-TiO₂ cores. As part of that effort, we prepared electrostatically associated POM ligands on TiO₂. These and numerous other lines of evidence were needed to convince us that (in the new structures) the POM ligands were **covalently** attached. That situation is fundamentally and conceptually different from the electrostatically stabilized TiO₂ cores we ourselves prepared (see the above figure), and from the electrostatically stabilized TiO₂ and Fe₂O₃ NCs prepared by Talapin (i.e., *ACS Nano*, **2014**). In that article, Talapin writes, "... in addition to Coulombic interactions between charged NC surface and POM molecules, van der Waals forces can additionally strengthen bonding between NCs and POM." A graphical illustration in Figure

- α- S15 of Talapin's 2014 article makes it very clear that they (correctly) refer to electrostatically stabilized colloidal NCs. **Below-left** is a copy of that illustration, and **to its right** is a figure from our present manuscript.

Notably, the POMs in our present manuscript are **covalently attached**, giving a type of **coordination complex**, a situation conceptually distinct (and unique) from the **electrostatically stabilized** structures reported by Talapin. This is why we refer to them as occupying a unique position between macroanions and traditional (electrostatically stabilized) colloidal NCs. **This is a conceptual advance with tremendous implications for the use of metal-oxide nanocrystals in catalysis.**

Reproduced from Figure S15 of Talapin's **2014** *ACS Nano* article, showing weakly **(electrostatically) associated** POM anions.

From **Figure 1** of our present manuscript showing **covalently attached** POM ligands.

This leads directly to the second fundamental advance in our manuscript. The covalent attachment of the POM ligands renders our new hematite structures (1) **water-soluble** and **substitutionally inert**, as well as **inherently stable to oxidation and hydrolysis**. Notably, they are stable in water from pH 2 to 8! These features make it

possible to use them as soluble analogs of **molecular** visible-light driven water-oxidation catalysts (WOCs).

Attesting to the simple logic of this approach, the new water-soluble catalyst is operated for an entire week (7 days; see Figure 4d of the revised manuscript) **with no decrease in activity whatsoever, a longer time under turnover conditions in water than ever documented for any molecular water-oxidation catalyst.**

Point-by-point responses to reviewers 1, 2 and 3

In this regard, reviewer 2 notes that "water oxidation has already been conducted using Fe₂O₃ nanocrystals decorated with [P₂Mo₁₈O₆₂]⁶⁻ ligands" (i.e., by Talapin in the cited 2014 *ACS Nano* article). In the Experimental section of that work the authors explain that, "The NC films were prepared at the surface of working electrodes by drop casting." Notably, the deposition of iron-oxide nanoparticles, nanocrystals and thin iron-oxide films, and their use as electrocatalysts and / or photoelectrocatalysts has a long history. Talapin's results are in line with those reported in numerous articles over the past two to three decades (too numerous to cite here).

Regarding stability under turnover conditions, reviewer 2 refers to a Wells-Dawson anion ([P₂Mo₁₈O₆₂]⁶⁻) mentioned in the abstract to Talapin's article, but in the Experimental Section and Supporting Information, the Keggin dodecamolybdophosphate anion is listed. The formula in the abstract seems to be incorrect. This is important because Talapin writes that, "For electrochemical measurements in aqueous solutions, the pH was buffered at 6.85 by a phosphate buffer..." **At this pH, PMo₁₂O₄₀³⁻ is extensively hydrolyzed to a mixture of products.** This serious problem is not addressed by Talapin.

By contrast, the main point of our work is that no decrease in reactivity of a *soluble* catalyst is observed even after 3 days (and now in the revised manuscript, after *one week*) under turnover conditions at pH 8. In short, electrocatalysis by iron-oxide films (with or without POMs) has little relation to the context and goals that define the conceptual framework behind the work reported in our manuscript.

To summarize, the conceptually new aspects of our work concern both **structure** and **reactivity**:

Structure: We report the covalent attachment of POM to iron-oxide cores, to give water-soluble complexes that occupy a unique position between molecular macroanions and (traditional) electrostatically stabilized colloidal iron oxide.

The structure we report is fundamentally and conceptually different from the electrostatically stabilized NCs reported by Talapin.

Reactivity: This new structure is the basis for the unprecedented stability of our catalyst under turnover conditions in water. I.e., we report remarkable (thermodynamic) stability—*by design*—of a soluble WOC under turnover conditions in water.

This visible-light driven solution-state catalysis is entirely distinct from Talapin's use of deposited iron oxide NCs in electrocatalytic water oxidation.

We hope that the above discussion helps reviewer 2 to appreciate the conceptual novelty of our work. In the revised manuscript, we provide a less detailed discussion, opting instead for an objective, informative and scholarly addition to the text. In it, we indicate what is conceptually new in our work, but do so in a manner that is respectful of Talapin's seminal contributions. That text is reproduced here:

Changes made in revision: The following discussion (including citations of six additional articles—see below) has been added to the main text of the revised manuscript:

"This covalent attachment of POM cluster-anions is fundamentally distinct from the use of POMs to electrostatically stabilize colloidal metal-oxide NCs. Notably, in a series of seminal papers, Murray,[1] Milleron and Helms[2] and Talapin,[3] used NOBF₄, Me₃OBF₄ and Ph₃CBF₄, respectively, to replace organic protecting ligands on metal-oxide NCs by electrostatically associated BF₄⁻ anions.[4] Milleron[5,6] and Talapin[3] then replaced the weakly bound BF₄⁻ anions by hexaniobate and other POMs, respectively, giving clear solutions of NCs electrostatically stabilized by the POM anions. Having removed the more tightly bound organic ligands, Milleron used hexaniobate-stabilized Sn-doped In₂O₃ (ITO) NCs as building blocks for tunable nanocrystal-in-glass composites,[5,6] while Talapin showed that films prepared by depositing POM-stabilized Fe₂O₃ on ITO electrodes were more effective electrocatalysts for water oxidation than analogous ones prepared from organic-ligand protected NCs.[3]

In the present work, the covalent attachment of POM ligands to the hematite cores of **1** gives *substitutionally inert* structures that occupy a unique position at the interface between molecular iron-oxide

Point-by-point responses to reviewers 1, 2 and 3

clusters and electrostatically-stabilized colloidal iron-oxide NCs. We now show that, when used as a soluble photocatalyst for visible-light driven water oxidation, **1** is inherently stable under turnover conditions in water."

38 Dong, A.; Ye, X.; Chen, J.; Kang, Y.; Gordon, T.; Kikkawa, J. M.; Murray, C. B. A generalized ligand-exchange strategy enabling sequential surface functionalization of colloidal nanocrystals. *J. Am. Chem. Soc.* **2011**, *133*, 998–1006.

39 Rosen, E. L.; Buonsanti, R.; Llordes, A.; Sawvel, A. M.; Milliron, D. J.; Helms, B. A. Exceptionally mild reactive stripping of native ligands from nanocrystal surfaces by using Meerwein's salt. *Angew. Chem., Int. Ed.* **2011**, *51*, 684–689.

40 Huang, J.; Liu, W.; Dolzhenkov, D. S.; Protesescu, L.; Kovalenko, M. V.; Koo, B.; Chattopadhyay, S.; Shenchenko, E. V.; Talapin, D. V. Surface functionalization of semiconductor and oxide nanocrystals with small inorganic oxoanions (PO_4^{3-} , MoO_4^{2-}) and polyoxometalate ligands. *ACS Nano*, **2014**, *8*, 9388-9402.

41 Nag, A.; Kovalenko, M. V.; Lee, J. S.; Liu, W.; Spokoyny, B.; Talapin, D. V. Metal-free Inorganic Ligands for Colloidal Nanocrystals: S^{2-} , HS^- , Se^{2-} , HSe^- , Te^{2-} , HTe^- , TeS_3^{2-} , OH^- , and NH_2^- as Surface Ligands. *J. Am. Chem. Soc.* **2011**, *133*, 10612–10620.

42 Llordes, A.; Hammack, A. T.; Buonsanti, R.; Tangirala, R.; Aloni, S.; Helms, B. A.; Milliron, D. J. Polyoxometalates and colloidal nanocrystals as building blocks for metal oxide nanocomposite films. *J. Mater. Chem.* **2011**, *21*, 11631–11638.

43 Llordes, A.; Garcia, G.; Gazquez, J.; Milliron, D. J. Tunable near-infrared and visible-light transmittance in nanocrystal-in-glass composites. *Nature*, **2013**, *500*, 323–326.

Point-by-point responses to reviewers 1, 2 and 3

Reviewer 3

Reviewer 3 summarizes the new findings and comments on their significance as follows: "...the authors described a method for covalently attaching polyoxotungstate to hematite cores, giving stable anionic structures (complex **1**) that are highly resistant to aggregation and stable to oxidation and hydrolysis... Complex **1** is shown to be a highly stable catalyst for visible-light induced water oxidation The use of the oxidation resistant polyoxotungstate to bind to hematite so to prevent aggregation is a novel approach to the design of stable water oxidation catalysts. This work should be of interest to researchers in the field of water oxidation, it could provide a new research direction to the general area of photocatalysis."

Regarding technical aspects, **reviewer 3** writes, "This paper is expertly written, the catalyst has been rigorously characterized by various techniques, including electron microscopy, mass spectrometry and infrared spectroscopy. The experimental part is clearly written. Extensive experimental data are also provided in Supporting Information."

Reviewer 3 clearly understands the **context**, **novelty** and **significance** of the work, and we appreciate his/her thoughtful suggestions for revision. Both points of science raised in his comments have been fully addressed.

We particularly wish to thank this reviewer for suggesting we push the catalyst further to better demonstrate its inherent stability under turnover conditions in water. As shown below (and in the revised manuscript), we've now increased the reaction time under turnover conditions from **3 days (3200 turnovers)** to an entire week (**7 days and 7600 turnovers**), again, with **no evidence for any decrease in reactivity**. We are delighted to document this, and indeed, it does make a stronger case for the main point of the article, i.e., that covalent attachment of POM ligands can be used to obtain a soluble visible-light driven water oxidation catalyst that is inherently stable to oxidative and hydrolytic degradation under turnover conditions in water.

Comment 1. "The authors seemed to claim that the functions of the Fe(III) substituted polyoxotungstate anions are to solubilize the hematite and prevent aggregation. However, since a number of molecular iron complexes are known to be efficient water oxidation catalysts, it is possible that in this system part of the oxygen evolved may arise from catalysis by Fe(III)-POM."

Response: This is indeed an important point. Although we addressed it in the original manuscript, we did so in a rather terse fashion (in Table 1 and a very brief discussion in the text). In response, we've tried to make our data on this issue more prominent. But first, let me explain what we did experimentally.

Please see **entries 5, 6 and 7 in Table 1**. **Entry 1** is the dark reaction between periodate and complex **1** (the POM-ligated hybrid). One function of this entry was to rule out a dark reaction between periodate and oxo-bridged Fe(III) atom in the covalently attached POM ligands at the hematite surface. Notably, no O₂ was observed. **Entry 6** is the dark reaction between periodate the molecular complex, $[(\alpha\text{-PW}_{11}\text{O}_{39}\text{Fe})_2\text{-}\mu_2\text{-O}]^{10-}$ (**2**). This was defined earlier in the text (before Figure 3). We probably should have written out the formula in Table 1 (we've now done that in the revised manuscript). This control experiment gave a trace amount of O₂ (ca. 1% of that obtained using the POM-complexed hematite) ruling out any significant contribution to oxygen evolution arising from catalysis by the Fe(III) POM ligands. In **entry 7**, the reaction of periodate and **2** was carried out under visible-light irradiation. A similar trace amount of O₂ was obtained. On this basis, we attribute the trace amounts of O₂ observed in both cases (in the dark and under visible light) to the known, "dark" reaction of periodate with molecular Fe complexes. This not only rules out catalysis by the Fe(III) POM, but more specifically, rules out oxo transfer from periodate to the Fe(III) POM. Based on the literature, we clearly needed to rule that out, and have done so.

The text in the original manuscript (in the fourth paragraph after Table 1; Page 11) was as follows:

"Secondly, oxo-transfer mechanisms invariably involve *thermal* (dark) *reactions* of molecular Fe, Ru and Ir complexes.^{46-48,51} By contrast, dark reactions of IO₄⁻ with **1** gave no O₂ (entry 5), while the dark reaction of IO₄⁻ with $\{[\alpha\text{-PW}_{11}\text{O}_{39}\text{Fe}]_2\text{O}\}^{10-}$ (**2**) gave only traces of O₂ (entry 6). The same reaction in visible light (entry 7) also gave traces of O₂, probably due to the dark reaction."

Point-by-point responses to reviewers 1, 2 and 3

Changes made: To make it clear that we addressed this issue by carrying out control experiments, we've made a few changes. I hope these are sufficient, and that reviewer 3 will be pleased that the science is correct, even if length limitations make it difficult to discuss the issue in more detail. The changes are as follows:

- 1) The formula for the molecular Fe(III) POM complex is now listed in Table 1 (see Table 1 in the revised manuscript).
- 2) The purpose of the control experiments in entries 6 and 7 is now stated in the footnote to Table 1 (see revised manuscript).
- 3) The discussion of these control experiments in the text (quoted above) has been modified. The revised text reads as follows (yellow-highlighted text is new):

"Secondly, oxo-transfer mechanisms (noted above) invariably involve *thermal* (dark) *reactions* of molecular Fe, Ru and Ir complexes.^{46-48,51} By contrast, dark reactions of IO_4^- with **1** gave no O_2 (entry 5), while the dark reaction of IO_4^- with $\{[\alpha\text{-PW}_{11}\text{O}_{39}\text{Fe}]_2\text{O}\}^{10-}$ (**2**) gave only traces of O_2 (entry 6). The same reaction in visible light (entry 7) also gave traces of O_2 , probably due to the dark reaction. These findings definitively rule out catalysis by the $[\alpha\text{-PW}_{11}\text{O}_{39}\text{Fe}^{\text{III}}]^{4-}\text{-O}^-$ ligands bound via oxo-linkages to the $\alpha\text{-Fe}_2\text{O}_3$ surface, including via oxo transfer from periodate to the POM-complexed Fe(III) atoms."

Comment 2. "The main drawback of this system is that efficient catalysis can also be achieved using periodate (persulfate is much less efficient) as the sacrificial oxidant. There are in fact, at least a couple of examples in the literature (Ref 11 and Chem. Commun. 2015, 51, 12189) of molecular iron complexes that readily catalyze thermal water oxidation by periodate, without the need of light and Fe_2O_3 semiconductor. The turnover numbers for the molecular catalysts are around 1000, which are just 3 times lower than the TON of 3200 for the much more complicated system reported in this paper. The authors claimed that their system is very stable and there is no decrease in catalytic activity after 3200 turnovers and 3 days of irradiation. Perhaps the authors can provide more convincing evidence on the stability of their catalysts, such as by doing catalysis for a longer period of time."

Response and changes made: We are grateful to referee 3 for encouraging us to operate the catalyst for a longer time to provide more convincing evidence of catalyst stability. **We've now carried out a photochemical reaction with 1 for seven days (an entire week).** The results are precisely what one would expect to see for the same catalyst (at the same concentration) operated for seven days, rather than for three. After three days, we observed 3200 turnovers. This value multiplied by 7/3 gives 7467 turnovers. **The experimental result (see new Figure reproduced below) is ca. 7600 turnovers.** As before, **this is achieved with no decrease in activity!**

We've now replaced the Fig. 4d (in main text) with our new data (shown below) for the seven-day reaction. Also, in the Methods section of the text, a paragraph is devoted to the specific conditions (oxidant loadings, etc.) needed to operate the catalyst for one week. In brief, to make up for consumed oxidant, additional IO_4^- was added during the reaction (to return to initial levels). To remove excess IO_3^- (product of the $2e^-$ reduction of IO_4^-) after days two, four and six, **1** was recycled by precipitation with NaCl, followed by centrifugation, re-dissolved in a fresh pH 8 solution of IO_4^- , and then returned to the visible-light reactor (see details provided in revised Methods section).

Point-by-point responses to reviewers 1, 2 and 3

Fig. 4. Visible-light driven water-oxidation by 1. d) Dioxygen produced by **1** ($5.4 \mu\text{M}$) as a function of time during seven days of reaction at pH 8. Oxygen production is plotted in mmol O₂ per g of $\alpha\text{-Fe}_2\text{O}_3$ in **1** (as routinely done for colloidal metal-oxide catalysts), and the turnover number (TON) is defined as mol O₂ / mol **1**, as is typical for molecular catalysis. The initial concentration of NaIO₄ was 20 mM. During the course of the one-week reaction, the solution was periodically charged with additional NaIO₄, and separated from accumulated iodate (IO₃⁻) (see the Methods section for details). The starting pH value and those after seven days, were 8.0 and 8.2, respectively; the buffering was provided by orthoperiodate, for which pK_{a2} and pK_{a3} are ca. 7.5 and 11, respectively.⁴³

Here is the newly added text from the **Methods** section, describing the above experiment:

Catalytic water oxidation for seven days under turnover conditions. A solution of **1** ($5.4 \mu\text{M}$) and NaIO₄ (20 mM) in 3 mL water at pH 8 (adjusted using 0.2 N KOH) was degassed as described above and irradiated with visible light ($\lambda > 420 \text{ nm}$) for one day, during which, amounts of O₂ in the headspace were quantified at regular intervals (see Fig. 4d). After one day, 6.4 mg solid NaIO₄ dissolved in 0.1 mL water was added to the solution (an additional 10 mM concentration of IO₄⁻) and the reaction continued for a second day. After day two, **1** was quantitatively separated from accumulated iodate (IO₃⁻; ca. 5.5 mM) by making the solution 0.5 M in NaCl, and isolating the salted-out catalyst by centrifugation. (A control experiment later carried out after recharging the supernatant solution with periodate, followed by irradiation, showed no activity (i.e., no O₂ in 8 h. This demonstrated that possibly unidentified soluble components were *not* responsible for the catalysis.) The pellet of **1** isolated after day two was then dissolved in 3 mL water containing freshly added periodate (20 mM), adjusted to pH 8, degassed, and irradiated for day three. (Similar separations of **1** from accumulated IO₃⁻ were repeated after days four and six.) After days three and five, additional 10 mM concentrations of IO₄⁻ were added. Turnover numbers were calculated based on the moles of O₂ produced per mole of **1**.

Closing comment. We are grateful to all three reviewers for their constructive comments and suggestions, in response to which we have: **1)** added a substantial amount of new data, **2)** revised the text to make it more clear, complete and accessible, and **3)** provided a new short section in the text (with new citations) to clarify what is conceptually new in our manuscript, in relation to the seminal work of Milleron and Talapin.

Reviewers 1 and 3 both note that the manuscript should be of interest to the field of water oxidation, and even for catalysis in general (reviewer 1). *All three reviewers* comment on the work's high scientific quality. We hope Reviewer 2 can now join with Reviewers 1 and 3 in recognizing the conceptually new approach taken here, and that all three reviewers can now recommend publication in *Nature Communications*.

Reviewers' Comments:

Reviewer #1:

Remarks to the Author:

The authors have addressed all the questions I raised in my original review. Most significantly, they have provided additional experimental evidence to support the claim of a covalent linkage between the hematite core and the POM ligand. Nicely, they have also shown that the catalyst activity lasts for several days with no evident decomposition.

I understand that the mechanistic analysis may be the subject of a different report, and the scheme provided in the current version of the ms is satisfactory for its clarity.

I am fully satisfied with the responses and the current version of the ms.

Reviewer #2:

Remarks to the Author:

It was interesting and meaningful for the author to re-visit the incremental development of related topics in the response/rebuttal letter in a pedagogical manner. We also went through the replies to other reviewers, and it was believed that the author properly addressed all the concerns and well-demonstrated the novelty of the current manuscript in comparison to the results in the literatures. We were finally convinced that this work was a fundamental progress in this field. Thus, we would like to suggest the formal publication of this work as its current status.

Reviewer #3:

Remarks to the Author:

In the opinion of this reviewer, the authors have made adequate revisions to their manuscript and it is now suitable for publication in Nature Communications.